# Influence of the Ti-TiN-(Y,Ti,Al)N Nanolayer Coating Deposition Process Parameters on Cutting Tool Oxidative Wear during Steel Turning

**DOI:** 10.3390/nano13233039

**Published:** 2023-11-28

**Authors:** Alexey Vereschaka, Catherine Sotova, Filipp Milovich, Anton Seleznev, Nikolay Sitnikov, Semen Shekhtman, Vladimir Pirogov, Natalia Baranova

**Affiliations:** 1Institute of Design and Technological Informatics of the Russian Academy of Sciences (IDTI RAS), Vadkovsky Lane 18a, 127055 Moscow, Russia; 2Department of High-Efficiency Machining Technologies, Moscow State University of Technology “STANKIN”, Vadkovsky Lane 3a, 127055 Moscow, Russia; e.sotova@stankin.ru (C.S.); a.seleznev@stankin.ru (A.S.); s.shekhtman@stankin.ru (S.S.); 3Materials Science and Metallurgy Shared Use Research and Development Center, National University of Science and Technology MISiS, Leninsky Prospect 4, 119049 Moscow, Russia; filippmilovich@mail.ru; 4The State Scientific Centre Keldysh Research Center, Onezhskaya St., 8, 125438 Moscow, Russia; sitnikov_nikolay@mail.ru; 5Department of Digital and Additive Technologies, Federal State Budget Educational Institution of Higher Education “MIREA—Russian Technological University”, Vernadsky Avenue, 78, 119454 Moscow, Russia; vpirogov@mai.ru (V.P.); baranova@mirea.ru (N.B.)

**Keywords:** nanolayer, coating, yttrium, wear resistance, oxidation, diffusion

## Abstract

Ti-TiN-(Y,Ti,Al)N coatings with a three-layer architecture (adhesive Ti layer, transition TiN layer, and wear-resistant (Y,Ti,Al)N layer) were studied. When depositing coatings, three arc current values of the yttrium cathode were used: 65, 85, and 105 A. The yttrium contents in the coatings were 30, 47, and 63 at. %, respectively. When turning 1045 steel, a coating with 30 at. % yttrium showed better wear resistance compared to a commercial (Ti,Cr,Al)N coating. The coating with 63 at. % yttrium did not show an increase in wear resistance compared to the uncoated sample. Nanolayers with a high yttrium content are oxidized more actively compared to nanolayers with a high titanium content. Phase analysis shows partial retention of the initial phases (Y,Ti,Al)N and (Ti,Y,Al)N during the formation of the Y_2_O_3_ oxide phase in the outer layers of the coating and the presence of only the initial phases in the deep layers. Coating nanolayers with high contents of aluminum and yttrium lose their original structure to a greater extent during oxidation compared to layers without aluminum.

## 1. Introduction

Modern mechanical engineering places increasingly stringent demands on structural materials. Traditional materials are often unable to meet these requirements, which predetermines the active development of materials with modifying coatings. In turn, increasing the performance properties of coatings is associated, among other things, with the search for new compositions. Nanolayer nitride coatings including yttrium are being actively researched and increasingly used in various fields of mechanical engineering. Let us look at some of the features of such coatings.

Yttrium nitride has several useful properties, but its tendency to decompose upon contact with water sharply reduces the scope of its practical application. A way out of this situation may be to create a coating in which layers with a high content of yttrium nitride alternate with layers that are not prone to reaction with water and thus perform an insulating role. Let us consider some properties of yttrium nitride. During the formation of yttrium nitride, a phase transition occurs in which the hcp structure of metallic yttrium is transformed into the fcc structure of yttrium nitride, but at low nitrogen pressure, the simultaneous presence of the hcp and fcc phases is observed [1,2]. At high pressure (more than 140 GPa), a transition from the fcc to the cP2 structure is observed, which is accompanied by a sharp compression in volume [3,4,5]. At a pressure of 170 GPa, the formation of a structure with a cF8 lattice, identical to the structure of high-pressure BaO, was observed [6]. In addition to YN nitride, other nitrides were also synthesized, such as Y_5_N_14_ (space group P4/mbm) [7]. A monolayer of nitride (t-YN) with strong anisotropy in mechanical and electronic properties was also obtained [8,9]. When contaminated with oxygen atoms during the synthesis process, yttrium nitride exhibits the properties of a metal [10]. The calculated hardness of YN is about 21 GPa, the elastic modulus E is about 290, and the B/G ratio is about 1.3, which characterizes YN as a brittle rather than ductile material [11]. By its nature, YN is an elastically stable and plastic material [12]. A useful property of yttrium nitride is also its high thermal stability [12]. As a rule, YN crystals with dimensions of several microns or even nanometers are formed. Moreover, under certain conditions, it was possible to obtain large crystals of yttrium nitride up to 200 μm in size [13].

The introduction of yttrium nitride into powder ferritic steel can significantly increase its strength and ductility [14]. The high hardness and melting point make YN suitable for applications such as magnetic recording and sensing. Yttrium nitride also has good potential for use in electronics and plasmonics as a refractory material with semiconductor properties [15,16,17,18]. Another area of potential application of yttrium nitride is optoelectronics due to the transparency and low reflectivity of YN [12,19] as well as unique properties such as negative refraction in all angles and anisotropic light propagation [20]. Finally, yttrium nitride has potential applications in photovoltaics and photocatalysis since two-dimensional tetragonal YN plates with nanometric thickness have high visible light absorption efficiency [21].

Possessing several useful features, YN is unstable in the air. Yttrium nitride crystals are transformed into yttrium oxide (Y_2_O_3_) after 2–4 h of exposure to air at room temperature [13]. This high oxidation tendency limits the use of YN. At the same time, alternating nanolayers based on YN with nanolayers of a different composition that are resistant to oxidation makes it possible to create coatings that combine the beneficial properties of YN and good oxidation resistance [22].

Coatings containing yttrium nitride, or yttrium in a solid solution, can be effectively used in various fields. The introduction of yttrium into coatings improves their properties. A study of the properties of the (Ti,Y)ON coating showed that at a yttrium content of less than 7.8 at. %, a single fcc phase of the solid solution is formed [23]. As the Y content increases, an amorphous yttrium oxide phase also forms. With increasing yttrium content, hardness and fracture toughness increased and the friction coefficient decreased [23,24,25]. The (Al,Cr,Y)N coating provides tool cutting properties that are superior to those of tools with a commercial (Ti,Al)N coating [24,25]. A more favorable wear pattern and better oxidation resistance were noted. A study of the wear resistance of tools with (Ti,Y,Al)N coating when turning steel showed a result close to the reference coating (Ti,Cr,Al)N. Compared to uncoated tools, the increase in wear resistance along the rake surface was 250–270%. A study of a nanolayer coating (Ti,Al,Cr)N/(Ti,Al,Y)N showed that the introduction of Y reduces the friction coefficient by 1.5 times at temperatures of 850–950° (temperatures characteristic of the cutting zone) [26]. In a (Cr,Al,Y)N/CrN nanolayer coating, the introduction of yttrium leads to a slight increase in the friction coefficient at temperatures below 300 °C; however, at higher temperatures, yttrium provides a noticeable decrease in the friction coefficient [27]. The presence of yttrium in the composition of the (Cr,Al)N coating can significantly improve its performance properties (Cr,Al,Y)N [28,29,30,31,32]. Y ions embedded in the coating make it possible to noticeably compact the microstructure of the coatings, especially the boundaries of columnar grains [33]. Thus, the effect of introducing yttrium is also the suppression of the processes of phase segregation and oxidation due to blocking diffusion paths along grain boundaries [32,33,34,35]. (Cr,Y,Nb)N coating has been effectively used as a thermal barrier to protect structural elements of turbines [36]. Studies were carried out on the properties of (Cr,Al,Y)N coatings with yttrium content from 0 to 10 at. % [34]. The wear resistance of coatings deteriorated with increasing Y content due to a decrease in hardness and an increase in the friction coefficient. Another problem was active cracking in the coating with a high yttrium content. When Y is introduced into the (Ti,Cr,Al)N coating, it can significantly increase oxidation resistance and reduce tool wear during cutting [37]. The introduction of yttrium into the coating composition can significantly reduce the resistance to temperature oxidation [38].

The (Ti,Y,Al)N coating has been studied in a number of works [39,40]. In particular, it was found that this coating has better resistance to high-temperature oxidation compared to the (Ti,Al)N coating [40]. As the Y content increases from 3 to 9 at. %, a structural change occurs from a single-phase cubic solid solution to a mixed cubic and wurtzite structure (at 5 at. % Y), and to a single-phase wurtzite structure at 9 at. % Y [40]. In this case, a decrease in grain size and a decrease in hardness are observed. The introduction of Y reduces the compressive stress and increases the hardness of the coating [41]. The addition of yttrium allows the formation of a dense nanostructure with small grains and a weak columnar morphology [42]. Due to this, surface roughness and the level of internal stresses are significantly reduced. As a result, the wear resistance of the coating increases resistance to cavitation and abrasive wear. The lattice parameter increased with increasing amounts of Y in the coating [38,42]. The presence of Y limits the diffusion of Fe through the coating during cutting. Yttrium oxide formed during heating segregates along the boundaries of oxide grains, limiting the diffusion of iron and oxygen in the coating [38]. The presence of Y in the coating can inhibit the growth of oxide grains, the formation of a layered oxide layer, and the compressive stresses associated with oxidation.

It is worth noting that as a rule, coatings containing no more than 10 at. % yttrium are considered. At the same time, there are results showing that in the presence of a nanolayer structure in which layers with a dominant content of yttrium nitride alternate with oxidation-resistant layers, coatings can work effectively [22]. Yttrium nitride tends to be destroyed upon contact with water (including that contained in the atmosphere). This is the reason for the low interest in coatings with high yttrium contents. At the same time, the creation of nanolayer structures with alternating barrier layers and layers based on yttrium nitride can expand the scope of application of coatings with a high content of yttrium nitride. Theoretically, such coatings can be used not only for cutting tools but also in other fields of activity in the production of semiconductors.

For this study, (Ti,Y,Al)N coatings with different yttrium contents were selected: 30, 45, and 65 at. %. In the coating under study, alternation of nanolayers based on YN with layers based on (Ti,Al)N should be ensured.

The Ti-TiN-(Y,Ti,Al)N coatings under study had a three-layer architecture [43,44,45,46] (Ti adhesion layer, TiN transition layer, and wear-resistant (Y,Ti,Al)N layer).

## 2. Materials and Methods

Coating deposition was carried out on a specialized VIT-2 installation (IDTI RAS—MSTU STANKIN, Moscow, Russia) [22,43,44,45,46,47,48,49]. During deposition, two varieties of the physical vapor deposition (PVD) method were combined: filtered cathodic vacuum arc deposition (FCVAD) [22,43,44,45,46,47] and controlled accelerated arc (CAA-PVD) technology [48,49]. FCVAD technology allows 95–99% separation of the plasma flow from the microparticle fraction. However, this method leads to a slight decrease in process productivity. FCVAD can be rationally used for the evaporation of light metals (primarily aluminum), during the evaporation of which via traditional methods a significant fraction of microparticles is formed. The CAA-PVD method has high productivity and at the same time allows a significant reduction in the number of microparticles. Thus, the Al cathode (99.80%) was installed on the evaporator of the FCVAD system, and the Y (99.98%) and Ti (99.75%) cathodes were installed on the evaporators of the CAA-PVD system. The cathode systems were installed at right angles to each other.

Carbide samples (WC + 5% TiC + 10% Co) with the SNUN shape in accordance with ISO 1832:2012 were studied. Before being placed in the chamber, the samples were washed in a special synthetic detergent with ultrasonic stimulation and then a final wash in purified ethyl alcohol.

The coating deposition process included the following steps:Preliminary thermal activation and etching in a flow of gas (argon) and metal (titanium) plasma.Deposition of coatings. The parameters for this process were selected based on the following analysis:

Changing the magnitude of the cathode arc current is an effective way to change the ratio of elements in the deposited coating. Having analyzed the available data on the influence of the arc current on the composition of coatings such as (Cr,Al,Si)N [50], (Ti,Al,Zr)N [51], (Zr,Nb)N [52], and (Nb,Y)N [53] using the similarity principle and based on the characteristics of the properties of metals, the arc current range was chosen from 40 to 140 A. As a result of the experiments, this range was narrowed to values from 65 to 105 A since values less than 65 A (in particular, 50 A) do not ensure stable arc combustion, and at values greater than 105 A (in particular, 125 A), a low-quality coating is deposited, which is not suitable for use as a wear-resistant coating. Thus, three values of yttrium cathode arc current were used: 65, 85, and 105 A. The corresponding coatings were designated Y65, Y85, and Y105, respectively. Other process parameters (titanium and aluminum cathode arc currents—110 and 160 A, respectively—gas (nitrogen) pressure 0.42 Pa, voltage on substrate −150 V, and turntable rotation speed, 0.7 rpm) remained constant for the three samples.

Wear resistance studies were carried out when turning 1045 steel on an ACU 500 MRD lathe (Sliven, Bulgaria) equipped with a ZMM CU500MRD variable-speed drive. Turning was carried out under dry cutting conditions. Cutting was carried out with geometric parameters: γ = −7°, α = 7°, λ = 0, r = 0.4 mm; in the following modes: feed (*f*) 0.1 mm/rev, depth (*a_p_*) 0.3 mm, speed (*v_c_*) = 400 m/min. The cutting speed is the limit for a tool with a reference (commercial) coating (Ti,Cr,Al)N. This speed was chosen in order to provide limiting cutting conditions, allowing us to study wear mechanisms (including oxidative processes). VB_max_ = 0.3 mm was adopted as a wear criterion. For each type of coating, 5 experiments were carried out, after which the average values of wear on the flank surface VB were determined.

To conduct studies of micro- and nano-objects, a scanning electron microscope (SEM) Carl Zeiss (Oberkochen, Germany) EVO 50, with EDX system X-Max—80 mm^2^ (OXFORD Instruments, Oxford, UK) was used in combination with a transmission electron microscope (TEM) JEM 2100 (JEOL Company, Tokyo, Japan). To study the elemental composition, TEM with EDX INCA Energy (OXFORD Instruments, Oxford, UK) was used. To obtain samples, the focused ion beam (FIB) method was used on a Strata 205 installation (Materials & Structural Analysis Division, Hillsboro, OR, USA).

The SV-500 nanoindentometer (Nanovea, Irvine, CA, USA), with a Berkovich indenter, was used to measure hardness and elastic modulus at a maximum load of 20 mN. The average values were determined based on the results of 20 measurements.

The scratch test was carried out on an SV-500 device (Nanovea, Irvine, CA, USA) in accordance with the ASTM C1624-05 method [54].

## 3. Results

### 3.1. Elemental and Phase Composition, Structure

The results of the elemental analysis of the coatings under study are presented in Figure 1. The nitrogen content is 50 ± 2 at. %; below is the content of only metals, excluding nitrogen. Thus, the total metal content is taken to be 100 at. %.

With an increase in the arc current of the yttrium cathode, the yttrium content increases (from 30.4 to 63.2 at. %), and there are simultaneous decreases in the titanium content (from 56.8 to 34.2 at. %) and aluminum content (from 7.1 to 3.0 at. %).

Analysis of the X-ray diffraction patterns (Figure 2) shows that it is difficult to see significant differences between samples using this method. Since a fairly thin coating is being studied—and moreover since the thickness of the coating on the Y105 samples varies considerably and the structure of this coating is non-uniform (see below)—significant errors may arise with this research method. However, X-ray diffraction analysis showed that in three coatings, in addition to the substrate phases (WC—COD 2300252, TiC—COD 9008747, Co—Crystallography Open Data-base (COD) 9011618), there are two phases of cubic solid solutions of nitrides: a (Ti,Y,Al)N (TiN—COD 1100033) phase with a = 4.24 Å and (Y,Ti,Al)N (YN—COD 9008768) with a = 4.88 Å. The difference between the two phases is that they are two different cubic solid solutions of nitrides which have the same unit cells (fcc) but different lattice parameters since in one, the Ti content dominates, and in the second, the Y content dominates. Due to the heterogeneity of the coatings and differences in thickness, it is impossible to draw a conclusion about the quantitative relationship of the phases from the diffractograms.

The total thickness of the coatings was 3.0 ± 0.3 μm.

A study of the nanolayer structure of coatings (Figure 3) shows the presence of a modulation period with alternating thin, light in contrast layers with a dominant Al content and thicker, dark in contrast layers with dominance of Ti and Y. In this case, layers with dominance of Ti and dominance of Y are impossible identify through contrast. It is worth noting that with increasing cathode arc current Y, a slight increase in the modulation period is observed. This may be due to an increase in deposition rate.

An analysis of the distribution of elements in coating nanolayers, studied using the Y85 coating as an example, shows that in nanolayers that are light in contrast, there is a higher (10–12 at. %) aluminum content (Figure 3b). In nanolayers that are darker in contrast, alternating dominance of titanium and yttrium is observed, while the boundaries of nanolayers rich in yttrium and rich in titanium are almost impossible to distinguish despite a noticeable difference in atomic mass.

### 3.2. Hardness, Elastic Modulus, Critical Fracture Load Values during Scratch Test

The hardness of the coatings noticeably decreases as the arc current of the yttrium cathode increases (Table 1). The critical fracture load for scratching also decreases as the arc current increases. From the point of view of the studied parameters, only Y65 and Y85 coatings have prospects for effective practical use.

### 3.3. Wear Resistance during Turning

Cutting tests when turning 1045 steel showed that only Y65 and Y85 coatings provide a significant increase in tool wear resistance (see Figure 4). The Y105 coated tool, considering the scatter of values, showed a wear rate almost identical to the uncoated tool. Compared to the reference coating (Ti,Cr,Al)N, the Y65 coating provided a slight reduction in wear rate. At the same time, tools coated with (Ti,Cr,Al)N and Y65 showed an identical tool life (360 s). Tools coated with Y85 provide some improvement in wear resistance, but in this indicator, this coating is inferior to both the Y65 coating and the reference coating, (Ti,Cr,Al)N. The average tool life of a tool with Y85 coating was 240 s, which is also lower than that of tools with (Ti,Cr,Al)N and Y65 coatings. Thus, only the Y65 coating has certain prospects for use in high-speed cutting conditions.

A study of the wear pattern on the rake surface of a coated tool after cutting 1045 steel (Figure 5) shows several differences. In particular, the Y65 coating is characterized by the formation of a smooth wear area of the coating, inclined relative to the surface of the substrate. For samples with Y85 and Y105 coatings, such an area is not observed; there is a clearly defined fracture boundary in the form of a step. On the surface of the Y85 and, especially, Y105 coatings, in the area adjacent to the cutting zone, the formation of damage spots is observed, presumably due to temperature oxidation. Analysis of element distribution maps (Figure 5d–f) shows that in these spots, there is a reduced yttrium content, while titanium is preserved—that is, it is the yttrium-rich coating layers that are partially destroyed. The features of oxidative processes will be studied in more detail below.

Comparison of cross-sections made in the wear area shows significant differences in the wear and failure of the coatings. If the Y65 coating is characterized by smooth wear without noticeable cracking or brittle fracture (Figure 6a), then the Y85 coating is characterized by the formation of delaminations between the nanolayers (Figure 6b). The Y105 coating is characterized by the formation of extensive delaminations combined with intense brittle fracture (Figure 6c). The thickness of the coatings in worn areas is different: if the Y65 coating has a thickness in the studied area of the order of 3.7 µm, then the thickness of the Y85 coating is about 4.9 µm, and that of Y105 is 5.2–6.0 µm. Such differences in coating thickness may also be due to the swelling of Y85 and, especially, Y105 coatings as a result of delamination and oxidative processes. The reason for such active and extended delaminations may be a high level of internal compressive stresses in combination with a relatively weak cohesive bond between nanolayers. It was previously found that with a high yttrium content, coatings become prone to embrittlement and cracking [33]. Another reason for the active destruction of coatings with a high yttrium content may be the low hardness of the yttrium-containing layers [33,41].

Let us examine in more detail the wear and destruction of the coatings under study when turning 1045 steel.

#### 3.3.1. Coating Y65

The results of the study of the distribution of elements on the rake surface of the Y65 coated tool after cutting 1045 steel are presented in Figure 7. The localization of the study area of the bottom row of maps is presented in Figure 5d–f. The most active oxidation is observed in the area of the cutting tip. Since high concentrations of titanium and aluminum are also noticeable in this area, the formation of oxides of these metals can be predicted. Since no increased concentration of yttrium is observed in this area, it can be assumed that yttrium oxides are absent or present in insignificant quantities. Since yttrium oxide is a fairly soft substance, it is quickly removed by the flow of chips and is not visible on element distribution maps. Other foci of active oxidation are associated with iron adherents. The distribution of yttrium and aluminum over the unworn part of the rake face is quite uniform. Individual spots with a high concentration of Y, up to 10 µm in size, may be associated with the presence of yttrium microparticles. The distribution of titanium is less uniform, except for possible microparticles (spots of increased concentration up to 10 µm in size). A concentration of titanium is observed along the boundary of the rake face (which, along with an increased oxygen content, may indicate the presence of titanium oxide in this area) as well as along the wear boundary of the coating (which may be due to the high titanium content in the adhesive sublayer, which appears during the process of wear of the coating).

A study of oxidation processes in the area of the Y65 coating adjacent to the wear boundary (Figure 8) shows that the outer layers of the coating, to a depth of 200 nm, show signs of active oxidation. In this case, nanolayers rich in yttrium are actively oxidized, but nanolayers dominated by Ti retain their original structure (see region A in Figure 8). Oxygen diffusion is observed to a depth of up to 300 nm. SAED analysis of the oxidized region of the coating (SAED 1, Figure 8) shows the formation of yttria Y_2_O_3_ while retaining the initial nitride phases. No titanium or aluminum oxides were detected in this area. SAED analysis of unoxidized coating layers at a depth of 300–500 nm from the surface (SAED 2, Figure 8) shows the presence of only initial nitride phases. A detailed analysis of the oxidized region A1 shows the formation of yttrium oxide Y_2_O_3_ while maintaining the nitride phase (Ti,Y,Al)N based on titanium nitride, but the nitride phase (Y,Ti,Al)N based on yttrium nitride is not identified (completely lost or preserved in insignificant quantity, insufficient for identification).

#### 3.3.2. Coating Y85

Analysis of element distribution maps on the worn rake face of a tool with Y85 coating shows that Y-containing coating layers begin to collapse not only directly in the cutting zone but also in the area adjacent to this zone. In Figure 9 (top row), it is observed that the yttrium content decreases sharply in the zone adjacent to the cutting edge, while the titanium content increases in this zone. Similar areas are observed along the wear boundary of the coating (Figure 9, bottom row). Thus, one can see the possible destruction of the yttrium-rich layers while maintaining the titanium-rich layers. Since in these areas there is no direct impact of the flow of cut material (and, accordingly, abrasive or adhesive-fatigue wear), it can be assumed that oxidative destruction of the coating occurs when exposed to high temperatures.

To test this assumption, consider the oxidative destruction of the Y85 coating (Figure 10). A coating defect which apparently arose during the deposition process is observed. This defect is possibly due to the unevenness of the substrate surface. The TiN transition layer is completely preserved, but the wear-resistant (Y,Ti,Al)N layer has undergone significant oxidative damage. The initial structure of (Y,Ti,Al)N was preserved in the internal nanolayers, within 2–3 modulation periods. The outer (Y,Ti,Al)N nanolayers underwent significant oxidation, which had a noticeable effect on their structure (Figure 10b–d). SAED analysis of the oxidized region shows both partial retention of the (Y,Ti,Al)N and (Ti,Y,Al)N phases and the formation of an Y_2_O_3_ oxide phase (Figure 10c). A more detailed examination of the structure of the oxidized region (Figure 10d) shows its heterogeneity. Along with deformed nanolayers, there are nanolayers that have retained their original structure. The swelling and deformation of the oxidized layers may be due to the fact that yttrium oxide has a density of 4.840 g/cm^3^, which is less than that of yttrium nitride (5.890 g/cm^3^) [55]. A comparison of the compositions of oxidized nanolayers and nanolayers with a preserved structure (Figure 10g) shows that a high aluminum content is observed in the oxidized layers, while no aluminum was found in layers with a preserved structure.

The question of the effect of the simultaneous presence of yttrium and aluminum in the composition of the alloy on oxidation resistance is quite complex. On the one hand, it was found that the introduction of both yttrium and aluminum into the alloy composition reduces the corrosion resistance compared to the introduction of yttrium alone [56]. The presence of yttrium in the alloy enhances the selective oxidation of Al [57], promoting the growth of α-Al_2_O_3_ [58]. Yttrium can have a noticeable effect on the phase transformation of metastable to α-alumina at the initial stage of oxidation [59]. It was found that the presence of yttrium ions at the grain boundaries of aluminum oxide has a significant effect on the oxidizing properties [60]. The oxidation rate decreases with the simultaneous addition of yttrium and aluminum due to the formation of a protective film of Al_2_O_3_ [61], while at the same time, the selective oxidation of Al is accelerated [62]. There are several studies showing the positive effect of introducing yttrium into alloys containing aluminum on their oxidation resistance [63,64,65,66,67,68], but such an increase in oxidation resistance is associated with the formation of a dense protective film of Al_2_O_3_ instead of a loose oxide layer and not with an increase in the direct oxidative properties of the alloy. Thus, if we are discussing conditions under which a protective film of Al_2_O_3_ is not formed (due to insufficient aluminum content) or this film is destroyed due to active deformation, it can be assumed that the combination of yttrium with aluminum leads to an acceleration of oxidative processes.

The most detailed oxidized region of the Y85 coating is shown in Figure 10f. In the oxidized mass, fragmented nanolayers with a partially preserved structure are observed. The thickness of such fragments can be only 8 nm. There is a “dissolution” (gradual oxidation) of the remaining fragments of the original coating structure. Active oxidation of yttrium (simultaneously high yttrium and oxygen content) is observed at points 1 and 7. At the same time, at point 2, which has a dominant titanium content, a decrease in oxygen content is observed, which may indicate a lower degree of oxidation.

#### 3.3.3. Coating Y105

Analysis of element distribution maps on the rake surface of a Y105 coated tool (Figure 11) shows active oxidative destruction of yttrium-rich nanolayers in the area adjacent to the cutting zone. In this case, areas with high yttrium content practically coincide with areas rich in aluminum. Comparison of elemental contents in a representative fragment of the wear boundary region of the coating on the rake surface of the tool (Figure 11b) shows that the region with high oxygen content coincides with regions rich in yttrium and aluminum and does not coincide with the region rich in titanium. This fact can serve as additional confirmation of the increased susceptibility to oxidation of regions with high yttrium and aluminum content.

The oxidative destruction of the Y105 coating is similar to the Y85 coating. Swelling of the oxidized layers occurs due to a decrease in the density of the substance, due to which a noticeable deformation of the preserved nanostructure occurs (Figure 12a–c). A kind of honeycomb structure is formed in which stronger preserved fragments of nanolayers restrain the expansion of oxidized zones. Simultaneous and uniform oxidation of the coating layers is not observed. Depending on the composition, some areas are actively oxidized and lose their nanostructure, while others retain their original structure. Analysis of the elemental composition of various nanolayers shows (Figure 12d) that layers with a high aluminum content are actively oxidized and lose their initial structure. Nanolayers, even those with high yttrium content but low aluminum content, are oxidized but at the same time retain a dense structure. Since there are areas with a deformed structure but a relatively low oxygen content, it can be assumed that along with oxidation processes in layers with a high Al content, when exposed to high temperature, spinodal decomposition can take place [69,70,71,72,73].

Analysis of the distribution of elements in the coating (Figure 12d) shows higher oxygen content in areas with high yttrium content, indicating more active oxidation of these nanolayers. The oxidized regions have a predominantly amorphous structure, but fragments with a crystalline structure may also be present.

SAED analysis shows, similarly to that observed above, partial retention of the (Y,Ti,Al)N and (Ti,Y,Al)N phases during the formation of the Y_2_O_3_ oxide phase in the outer layers of the coating and the presence of the initial phases (Y,Ti,Al)N and (Ti,Y,Al)N in deep layers (Figure 12f).

Thus, the tendency of yttrium nitride to oxidize in an air environment, studied earlier [13], is confirmed. Such oxidation is actively manifested in the coating containing Y105, containing 63 at. % yttrium, and, partly, in the Y85 coating containing 47 at. % yttrium. However, for the Y65 coating containing 30 at. % yttrium, such oxidation is much less pronounced. As noted earlier, alternating nanolayers with high and low yttrium content in the coating structure makes it possible to significantly slow down the oxidation process [22]. In this case, nanolayers rich in titanium and aluminum protect nanolayers rich in yttrium from oxidation. It was previously noted that the introduction of yttrium into the coating composition can improve the cutting properties of the tool [24,25,28,29,30,31,32]. However, such conclusions were made when studying coatings with a relatively low yttrium content (no more than 10 at. %). The use of coatings with a higher yttrium content was considered unpromising due to the expected active oxidation at high temperatures. The results presented above show that coatings with a high yttrium content (30 and even 47 at. %) can be effective provided that they have a nanolayer structure that includes insulating layers that have a low tendency to oxidize. The use of such structures with a nanolayer structure can be effective not only when cutting materials but also, for example, when creating semiconductor structures, which is important considering the corresponding properties of yttrium nitride [15,16,17,18].

## 4. Conclusions

Ti-TiN-(Y,Ti,Al)N coatings having a three-layer architecture (adhesive Ti layer, transition TiN layer and wear-resistant (Y,Ti,Al)N layer) were studied. When depositing coatings, three arc current values of the yttrium cathode were used: 65, 85, and 105 A. The yttrium contents in the coatings were, respectively, 30, 47, and 63 at. %. Based on the results of the research, the following conclusions can be drawn:When turning 1045 steel, a coating with 30 at. % yttrium showed better wear resistance compared to a commercial (Ti,Cr,Al) N coating. The coating with 63 at. % yttrium did not show an increase in wear resistance compared to the uncoated sample.While the tool with a coating containing 30 at. % yttrium showed smooth wear without signs of active brittle fracture, other coatings under study were destroyed as a result of active cracking and oxidation.Nanolayers with a high yttrium content are oxidized more actively compared to nanolayers with a high titanium content. When the titanium content in nanolayers subject to oxidation is more than 50 at. %, observed within 10 at. % oxygen, with a content of more than 50 at. % yttrium oxygen content exceeding 25 at. %.SAED analysis shows partial retention of the (Y,Ti,Al)N and (Ti,Y,Al)N phases during the formation of the Y_2_O_3_ oxide phase in the outer layers of the coating and the presence of the initial phases (Y,Ti,Al)N and (Ti, Y,Al)N in deep layers.Nanolayers of coatings with a high aluminum content are more susceptible to oxidative attack with the loss of the original nanolayer structure. In addition to oxidation, spinodal decomposition processes can occur in these nanolayers. This issue requires additional study.

Combining nanolayers of materials that are prone and resistant to oxidation in the coating structure can significantly reduce the overall susceptibility of the coating to oxidation. Such nanolayer structures can be effectively used in various fields, in particular in the production of metal-cutting tools, in friction pairs and in the creation of semiconductors. By changing the arc current of the cathodes, it is possible to control both the composition of the nanolayers and their thickness, which makes it possible to effectively control the parameters of the structure and composition of the coatings.

## Figures and Tables

**Figure 1 nanomaterials-13-03039-f001:**
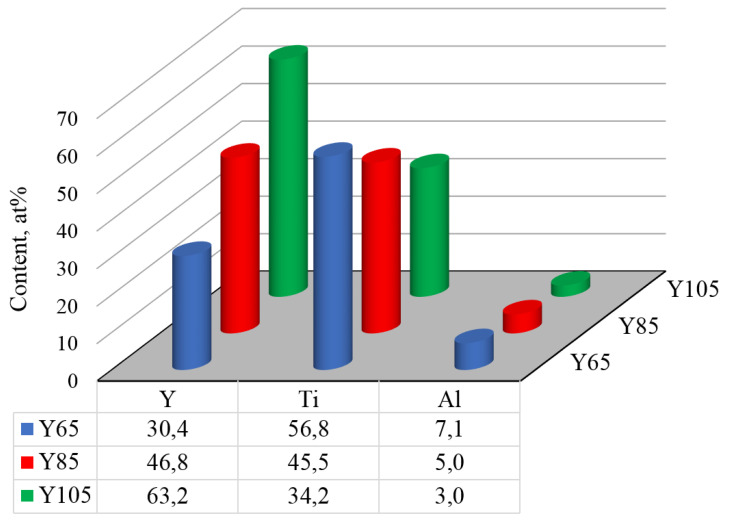
Elemental composition of the coatings under study. The total content of all metals, excluding nitrogen content, is taken as 100 at. %.

**Figure 2 nanomaterials-13-03039-f002:**
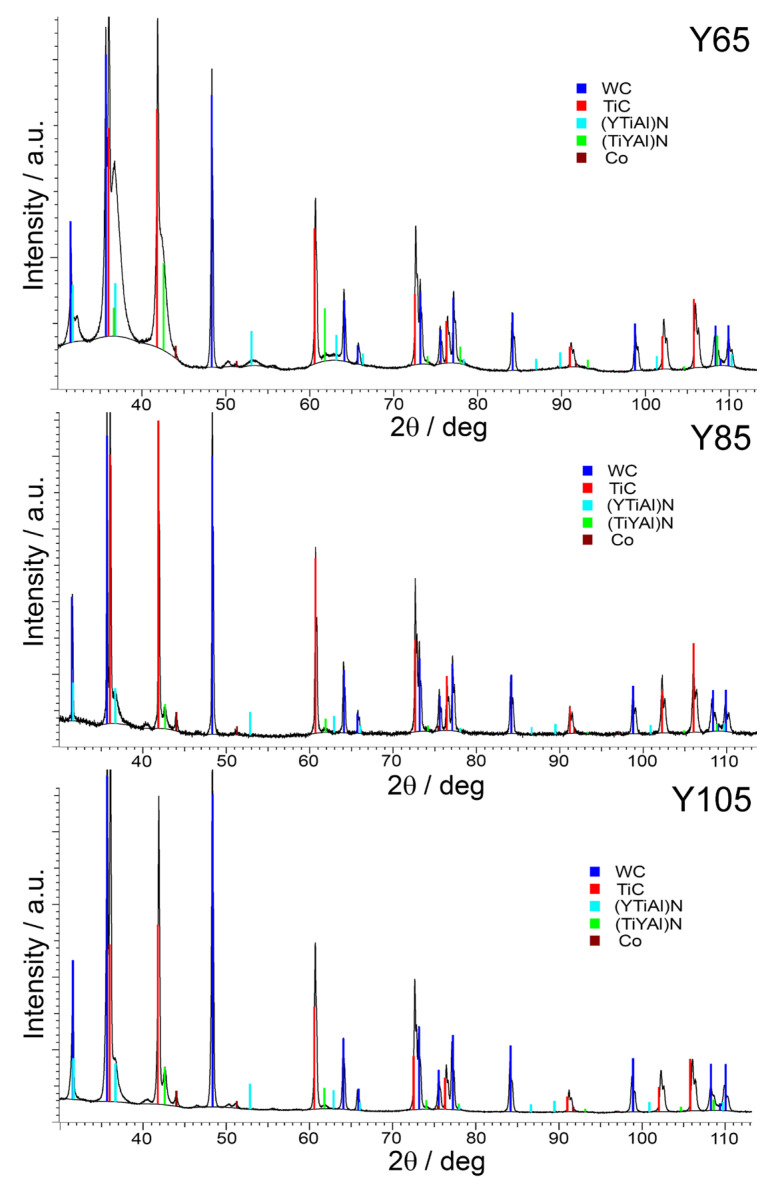
Phase composition of coatings studied via X-ray diffraction (XRD).

**Figure 3 nanomaterials-13-03039-f003:**
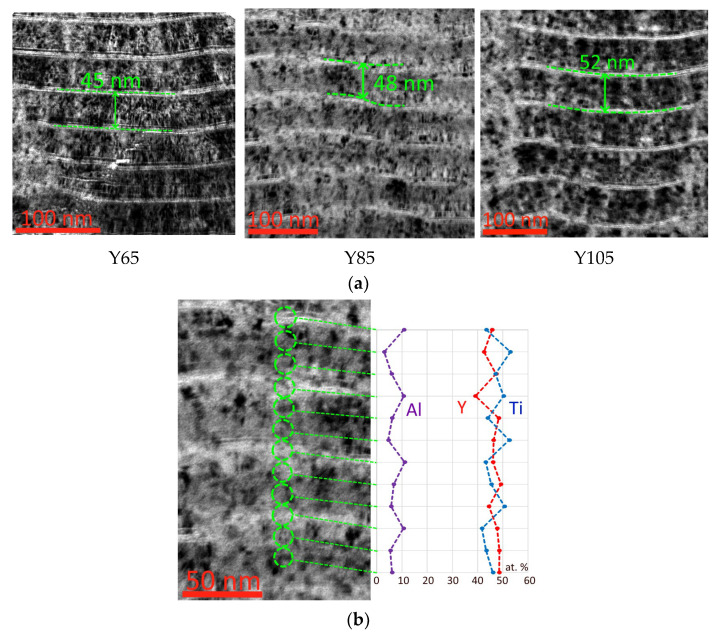
(**a**) Structure of the studied coatings, (**b**) distribution of elements in nanolayers of Y85 coating.

**Figure 4 nanomaterials-13-03039-f004:**
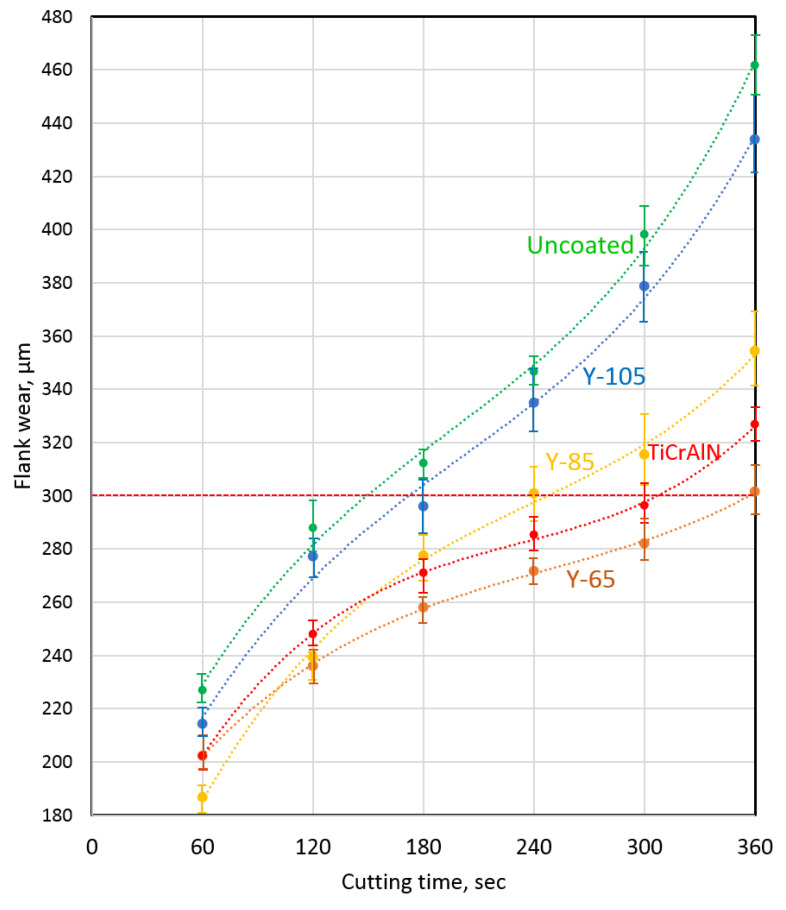
Study of the wear rate of cutting tools with coatings when turning 1045 steel.

**Figure 5 nanomaterials-13-03039-f005:**
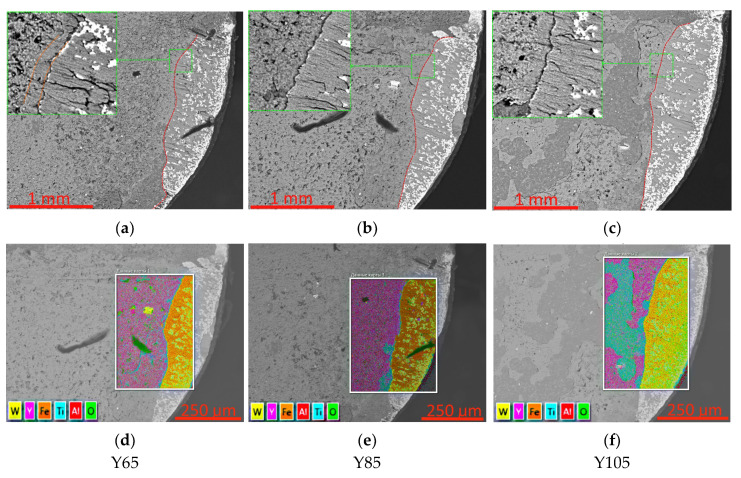
Condition of the worn rake face of a coated tool after 360 s cutting; (**a**–**c**) front surface view and pavement failure boundary, (**d**–**f**) location of mapping areas and general element distribution maps. The red dashed line indicates the boundary of the wear area.

**Figure 6 nanomaterials-13-03039-f006:**
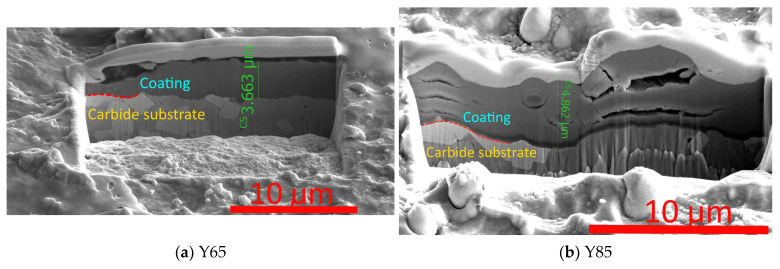
Transverse sections of coatings in the wear area (SEM).

**Figure 7 nanomaterials-13-03039-f007:**
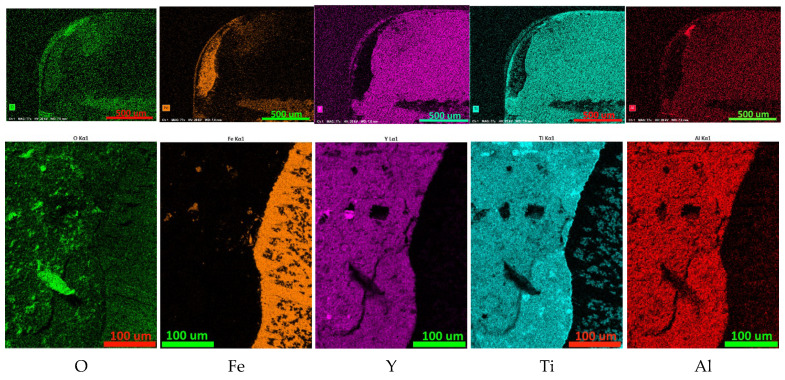
Mapping the distribution of elements on the rake face of a worn cutting tool coated with Y65. The top row is a general view of the worn part of the front surface; the bottom row is the distribution of elements in the area, the localization of which is presented in Figure 5d.

**Figure 8 nanomaterials-13-03039-f008:**
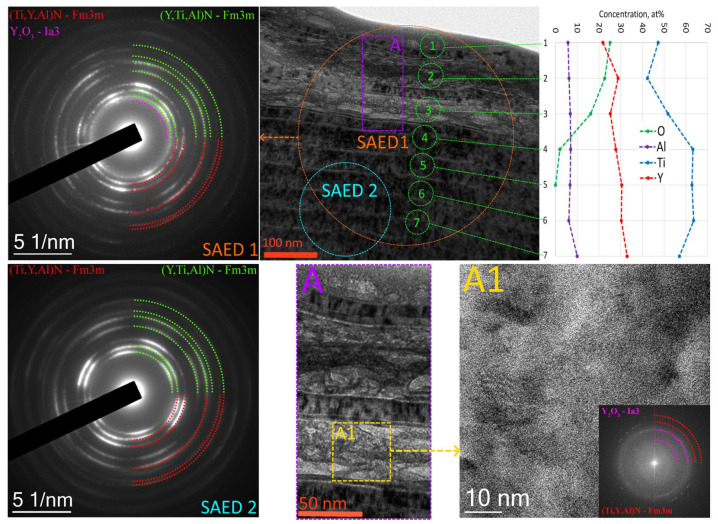
Study of oxidative processes in the area of the Y65 coating adjacent to the wear boundary.

**Figure 9 nanomaterials-13-03039-f009:**
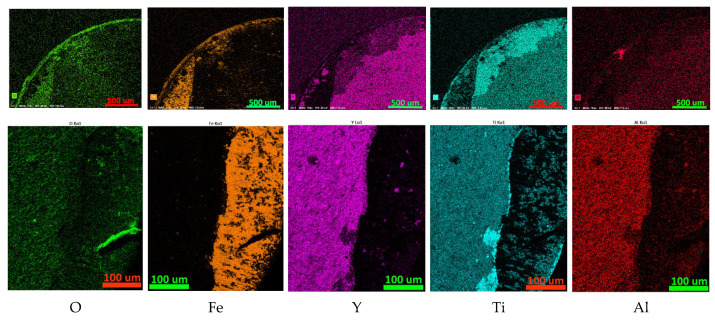
Mapping the distribution of elements on the rake face of a worn cutting tool coated with Y85. The **top** row is a general view of the worn part of the front surface, the **bottom** row is the distribution of elements in the area, the localization of which is presented in Figure 5e.

**Figure 10 nanomaterials-13-03039-f010:**
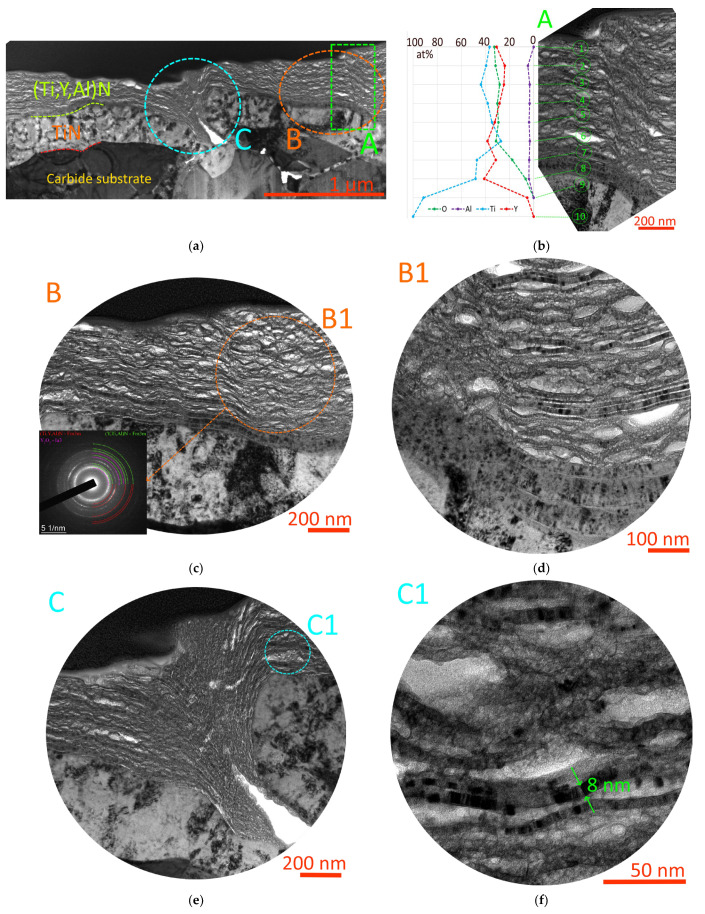
Study of the oxidative destruction of the Y85 coating during cutting. (**a**) General view of the lamella and localization of the areas of study; (**b**) analysis of the distribution of elements throughout the thickness of the coating layer; (**c**) study of the oxidized region, SAED analysis; (**d**) f the oxidative destruction of the wear-resistant layer; (**e**) oxidative destruction of the coating in the defect area; (**f**) fragments of the structure of the initial coating nanolayers; (**g**) study of the elemental composition of the coating nanolayers during the oxidation process.

**Figure 11 nanomaterials-13-03039-f011:**
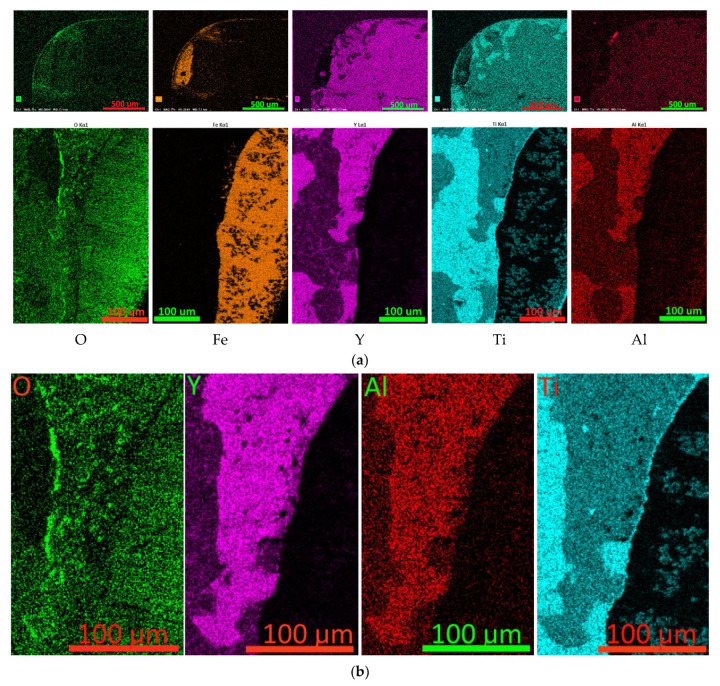
Mapping the distribution of elements on the rake face of a worn cutting tool coated with Y105. (**a**) Top row—general view of the worn part of the front surface, bottom row—distribution of elements in the area, the localization of which is presented in Figure 5e; (**b**) comparison of the elemental composition of a fragment of the coating wear boundary on the front surface.

**Figure 12 nanomaterials-13-03039-f012:**
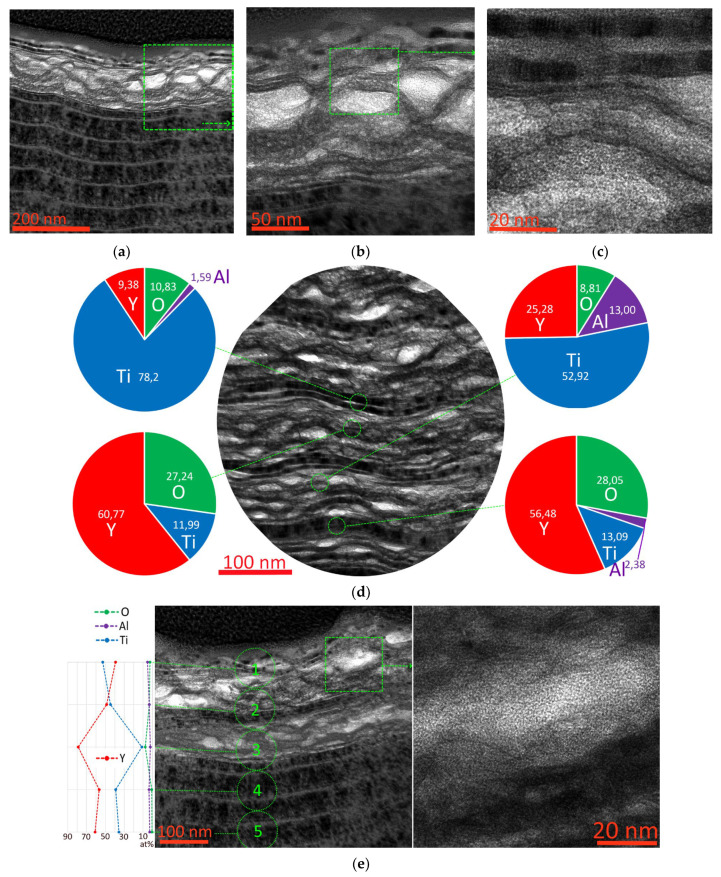
Study of the oxidative destruction of the Y105 coating during cutting. (**a**–**c**) Oxidized region of the coating at different image scales; (**d**) study of the elemental composition of coating nanolayers during the oxidation process; (**e**) analysis of the distribution of elements throughout the thickness of the coating layer; (**f**) study of the oxidized region, SAED analysis.

**Table 1 nanomaterials-13-03039-t001:** Results of the study of hardness, elastic modulus, and critical fracture load.

Coating	Hardness, GPa	Elasticity Modulus, GPa	Critical Failure Load LC2, N
Y65	27 ± 1.2	215 ± 47	38
Y85	24 ± 0.9	200 ± 62	24
Y105	16 ± 1.7	160 ± 67	6

## Data Availability

Data are contained within the article.

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
