# Peer review of "Influence of the Ti-TiN-(Y,Ti,Al)N Nanolayer Coating Deposition Process Parameters on Cutting Tool Oxidative Wear during Steel Turning"

_nanomaterials, 2023, doi:10.3390/nano13233039_

Round 1

Reviewer 1 Report

Comments and Suggestions for Authors

The influence of the coating preparation process on the wear resistance of cutting tools studied in this manuscript represents an interesting tribological study, and it may be suitable for publication in this journal. However, authors should attempt to address the following points raised. 

1.      In the Materials and Methods section, what does the geometric parameters γ, α, λ, and r refer to?

2.      The experimental details of XRD should be given. Similar, the experimental details of the scratch tests should be clarified rather than just mentioning the ASTM C1624-05 method.

3.      What is the loading, holding and unloading time in nanoindentation tests? And the typical load-displacement curves of each kind of coating should be given. Besides, what are the mechanical properties of uncoated cutting tools?

4.      Would it be possible for the authors to provide the optical micrograph, SEM image, or other similar comparison figures of the cutting tools before and after depositing the coatings?

5.      In Figure 4, the flank wear, name of the y-coordinate, refers more to the wear amount rather than the wear rate.

6.      What are the advantages of the coating preparation process used by the authors compared to existing traditional coating (like TiCN, TiAlN, c-BN, or polycrystalline diamond) preparation processes?

Comments on the Quality of English Language

There are some minor formatting errors in the manuscript, which the authors should check carefully.

Author Response

Reviewer 1 Comments and Suggestions for Authors

The influence of the coating preparation process on the wear resistance of cutting tools studied in this manuscript represents an interesting tribological study, and it may be suitable for publication in this journal. However, authors should attempt to address the following points raised.

A: The authors are grateful to the reviewer for valuable recommendations and comments that helped improve the quality of the manuscript.

  1. In the Materials and Methods section, what does the geometric parameters γ, α, λ, and r refer to?

A: Designations of the geometric parameters of the tool are presented in accordance with ISO 3002-1:1982 (Basic quantities in cutting and grinding. Part 1: Geometry of the active part of cutting tools). Text explanations have been added to the text of the article: rake angle γ = –7°, clearance angle α = 7°, tool cutting edge inclination λ = 0

  1. The experimental details of XRD should be given. Similar, the experimental details of the scratch tests should be clarified rather than just mentioning the ASTM C1624-05 method.

A: Descriptions of methods have been added to the Materials and Methods section

  1. What is the loading, holding and unloading time in nanoindentation tests? And the typical load-displacement curves of each kind of coating should be given. Besides, what are the mechanical properties of uncoated cutting tools?

The description of the method has been expanded. Unfortunately, the loading curves were not preserved. A large number of measurements were taken (at least 20), then the average value was determined. Measuring the hardness of coatings usually presents a number of difficulties, since:

- the surface of the coating has a relief associated with the cluster structure

- hardness is uneven and inconsistent, since there are boundaries of clusters, microparticles and other objects; moreover, in the center of a cluster the hardness is usually slightly higher than at the boundaries.

- the coating is a thin and very hard film on a relatively soft substrate (about 1000 HV) with a coating hardness of 2200 - 4000 HV. Thus, measurements are carried out at a minimum load to eliminate the influence of the substrate, but in this case there is a slight decrease in the measurement accuracy.

  1. Would it be possible for the authors to provide the optical micrograph, SEM image, or other similar comparison figures of the cutting tools before and after depositing the coatings?

A: Unfortunately, such a study has not been conducted. Figure 5 shows images of tools with coatings after wear. These images also contain an unworn part of the surface with coatings.

  1. In Figure 4, the flank wear, name of the y-coordinate, refers more to the wear amount rather than the wear rate.

A: The value of flank wear land VB was measured in accordance with ISO 3685. The caption and the corresponding text of the article were clarified.

  1. What are the advantages of the coating preparation process used by the authors compared to existing traditional coating (like TiCN, TiAlN, c-BN, or polycrystalline diamond) preparation processes?

A: The pre-treatment process before coating does not differ noticeably from the same process for TiCN, TiAlN, c-BN, or polycrystalline diamond. This process is described in the Materials and Methods section. The coating deposition process itself was carried out using filtered cathodic vacuum arc deposition (FCVAD) and Controlled Accelerated Arc (CAA-PVD) technology, which is described in the Materials and Methods section, which also provides references for a detailed description of these technologies.

Reviewer 2 Report

Comments and Suggestions for Authors

The article deals with an interesting and beneficial topic.

I have the following comments and recommendations:

1. What is meant by at%?

2. Fig. 1 has no reference in the text.

3. What does UNC mean in Fig. 4?

4. Fig. 5 - images (a, b, d, e) are flipped.

5. In Fig. 6 there is no designation a, b, c, d, e, but there is a reference to this in the text. It is necessary to unify similar situations throughout the text.

6. Labels 7, 9 and 11, always in the first line, are completely illegible (very, very small font).

7. I highly recommend changing the color scale for the labels in images 3, 6, 8, 9, 10 and 12, in some places, the chosen color makes these labels very, very illegible.

8. The word "nature" often appears in the text. I recommend using the word "area" in this context.

Author Response

Reviewer 2 Comments and Suggestions for Authors

The article deals with an interesting and beneficial topic.

A: The authors are grateful to the Reviewer for his valuable assistance in improving the manuscript.

I have the following comments and recommendations:

  1. What is meant by at%?

A: Corrected to at. %. atomic percent, which gives the percentage of one kind of atom relative to the total number of atoms

  1. Fig. 1 has no reference in the text.

A: Line 193: “The results of the elemental analysis of the coatings under study are presented in Figure 1.” Thus, Fig 1 has the corresponding reference.

  1. What does UNC mean in Fig. 4?

A: Corrected to Uncoated

  1. Fig. 5 - images (a, b, d, e) are flipped.

A: Unfortunately, I didn’t understand this remark - what does “are flipped” mean?

  1. In Fig. 6 there is no designation a, b, c, d, e, but there is a reference to this in the text. It is necessary to unify similar situations throughout the text.

A: Designations added

  1. Labels 7, 9 and 11, always in the first line, are completely illegible (very, very small font).

A: The above images have been modified for better viewing.

  1. I highly recommend changing the color scale for the labels in images 3, 6, 8, 9, 10 and 12, in some places, the chosen color makes these labels very, very illegible.

A: The authors ask that you trust that a variety of colors were tried and that those ultimately selected provide optimal visibility given the complex nature of the images.

  1. The word "nature" often appears in the text. I recommend using the word "area" in this context.

A: The word “nature” is basically redundant. It was removed without affecting the meaning of the phrases.

Reviewer 3 Report

Comments and Suggestions for Authors

Hard coating materials can effectively improve the performance of structural materials and have been applied in various fields of mechanical engineering. This work prepared Ti-TiN-(Y, Ti Al) N gradient coatings with high Y content, and studied the wear resistance and microstructure changes of coatings with different Y content after cutting wear experiments. The results have some novelty, but there are still many problems. It is not recommended to publish in the current state. The following are some specific comments.

1. In the introduction, many parts lack logic and need to be reorganized. the comprehensive generalization of the cited literature is insufficient. In the last part, the author should clearly present why it is necessary to study coatings with high Y content.

2. What is the thickness of the as-prepared coating? In Figure 3, it is recommended to use EDS technology for elemental analysis to determine which layers are rich in Y and which layers are rich in Ti ?

3. For the Y65 coating, on page 9, lines 275-279, the oxidation at the cutting tip is considered to be the oxidation of Ti and Al, without Y oxidation. However, on page 10, lines 296-297, it is claimed that nanolayers rich in yttrium are actively oxidized. Is there any relationship between the two? Is it because Y was destroyed by early oxidation that no oxidation of Y was observed at the tip? There is no explanation provided in the manuscript.

5. The colors for each element should be consistent in all images with element mappings. The rulers in images need to be enlarged to clearly present to the readers.

6. It is recommended to identify some detailed features in figures, such as the damage spots in Figure 5, the different layers and substrate in Figures 6 and 10. This can make reading easier for readers.

7. Page 15, line 405, Figure 12c should be changed to Figure 12d.

8. Y85 has better wear resistance than Y105. But by comparing Figures 10 and 12, it seems that  Y105 retains more original structure than Y85. This is confusing.

9. The structural changes of several coatings should be related to their wear resistance. This should be one of the focus of this work, but there seems to be no relevant evaluation in the manuscript. Are the elliptical structures in Figures 10 and 12 cavities? Y65 without such structures, has the best wear resistance. Y85 and Y105 with such structures, have worse wear resistance. Is there a relationship between wear resistance and this structure?

10. For Figure 6, it is claimed that differences in coating thickness are the results of delamination and oxidative processes. What is the reason for delamination? There is no explanation in the manuscript. In addition, Yttrium oxide can cause swelling, but only occurs in a few hundred nanometers thick oxide layer on the surface. However, the thicknesses of the coatings reach several micrometers. Obviously, the contribution of oxidation process to the thickening of the coating is negligible.

Comments on the Quality of English Language

 Quality of English language need to be improved

Author Response

Reviewer 3 Comments and Suggestions for Authors

Hard coating materials can effectively improve the performance of structural materials and have been applied in various fields of mechanical engineering. This work prepared Ti-TiN-(Y, Ti Al) N gradient coatings with high Y content, and studied the wear resistance and microstructure changes of coatings with different Y content after cutting wear experiments. The results have some novelty, but there are still many problems. It is not recommended to publish in the current state. The following are some specific comments.

A: The authors are grateful to the Reviewer for his valuable assistance in improving the manuscript.

  1. In the introduction, many parts lack logic and need to be reorganized. the comprehensive generalization of the cited literature is insufficient. In the last part, the author should clearly present why it is necessary to study coatings with high Y content.

A: The authors have made several changes to the Introduction to improve its quality and motivate research.

  1. What is the thickness of the as-prepared coating?

A: The (initial) thickness of the coatings was about 3 microns.

In Figure 3, it is recommended to use EDS technology for elemental analysis to determine which layers are rich in Y and which layers are rich in Ti ?

A: The results of a study of the distribution of elements in coating nanolayers (using the Y85 coating as an example) are added to Fig. 3, as well as the corresponding section in the text.

  1. For the Y65 coating, on page 9, lines 275-279, the oxidation at the cutting tip is considered to be the oxidation of Ti and Al, without Y oxidation. However, on page 10, lines 296-297, it is claimed that nanolayers rich in yttrium are actively oxidized. Is there any relationship between the two? Is it because Y was destroyed by early oxidation that no oxidation of Y was observed at the tip? There is no explanation provided in the manuscript.

A: The reviewer is absolutely right. There is, of course, no contradiction here. Since yttrium oxide is a fairly soft substance, it is quickly removed by the flow of chips and is not visible on element distribution maps. At the same time, the yttrium oxide layer is clearly visible on the lamellae examined by TEM, since the area adjacent to the cutting zone is being studied and the TEM method is much more accurate compared to the previously presented mapping.

  1. The colors for each element should be consistent in all images with element mappings.

A: Element cards are unified by color.

The rulers in images need to be enlarged to clearly present to the readers.

A: The scale bars were drawn on a larger scale.

  1. It is recommended to identify some detailed features in figures, such as the damage spots in Figure 5, the different layers and substrate in Figures 6 and 10. This can make reading easier for readers.

A: The figures have been modified and supplemented in accordance with the Reviewer's recommendations.

  1. Page 15, line 405, Figure 12c should be changed to Figure 12d.

A: Typo corrected

  1. Y85 has better wear resistance than Y105. But by comparing Figures 10 and 12, it seems that Y105 retains more original structure than Y85. This is confusing.

A: Unfortunately, when making a lamella, it is impossible to predict in advance the wear pattern in one area or another. The wear rate of the tool is determined by periodically measuring the wear chamfer on the rear surface of the cutting tool. The lamella was cut out from the front surface based on the analysis of element distribution maps. However, the cutting process is a very complex phenomenon and the temperature distribution depends on a number of factors. Thus, by comparing the degree of oxidation in equivalent zones, one can obtain paradoxical results, in which oxidation is more active on a tool that has generally better wear resistance. As you can see, entire areas of the yttrium-rich layer have been lost on the Y105 coating. Accordingly, areas with a preserved yttrium-rich layer were investigated. These areas may have a lower degree of oxidation.

  1. The structural changes of several coatings should be related to their wear resistance. This should be one of the focus of this work, but there seems to be no relevant evaluation in the manuscript. Are the elliptical structures in Figures 10 and 12 cavities? Y65 without such structures, has the best wear resistance. Y85 and Y105 with such structures, have worse wear resistance. Is there a relationship between wear resistance and this structure?

A: As far as we can understand, the elliptical regions are not cavities, but are yttrium oxide, which has a noticeably lower density compared to yttrium nitride. Thus, when the nitride transforms into oxide, the material swells and the structure shown in the images is formed.

  1. For Figure 6, it is claimed that differences in coating thickness are the results of delamination and oxidative processes. What is the reason for delamination? There is no explanation in the manuscript. In addition, Yttrium oxide can cause swelling, but only occurs in a few hundred nanometers thick oxide layer on the surface. However, the thicknesses of the coatings reach several micrometers. Obviously, the contribution of oxidation process to the thickening of the coating is negligible.

A: The authors tried to offer a possible explanation for the active destruction of the coating with a high yttrium content. The reason for such active and extended delaminations may be a high level of internal compressive stresses in combination with a relatively weak cohesive bond between nanolayers. It was previously found that with a high yttrium content, coatings become prone to embrittlement and cracking [34]. Another reason for the active destruction of coatings with a high yttrium content may be the low hardness of the yttrium-containing layers [34,42].

Comments on the Quality of English Language

 Quality of English language need to be improved

A: The text has been edited using professional proofreading (Cambridge Proofreading).

Reviewer 4 Report

Comments and Suggestions for Authors

Authors are strongly advised to reduce their self-citations (12 out of 84 is too many) and to discuss the nanostructures much more in detail. Authors at least try to obtain DF-TEM images to examine the dispersion and average sizes of dispersed nanoparticles. In addition, the same color sets should be used for facile comparison of elemental distributions by EDS of the general rows, etc. Furthermore, I am not entirely sure of the authors presenting circular fields of view in Fig.10. EDS maps by TEM at higher magnifications are desirable.

Comments on the Quality of English Language

Not English, but the way authors describe results should be greatly improved. Their results are too poor quality anyway.

Author Response

Reviewer 4 Comments and Suggestions for Authors

A: The authors are grateful to the Reviewer for his valuable assistance in improving the manuscript.

Authors are strongly advised to reduce their self-citations (12 out of 84 is too many) and to discuss the nanostructures much more in detail.

A: Number of self-citations reduced

Authors at least try to obtain DF-TEM images to examine the dispersion and average sizes of dispersed nanoparticles.

A: Unfortunately, the methods available to the authors do not allow us to obtain a quantitative grain size in nanolayer structures; we can only talk about a qualitative assessment, which, in our opinion, is also on the verge of reliability.

In addition, the same color sets should be used for facile comparison of elemental distributions by EDS of the general rows, etc.

A: Element cards are unified by color.

Furthermore, I am not entirely sure of the authors presenting circular fields of view in Fig.10. EDS maps by TEM at higher magnifications are desirable.

A: Figure 10g has been modified and expanded. Unfortunately, it is impossible to reliably measure the composition of regions smaller than 10 nm in diameter using available tools. Thus, mapping such small areas will not give the desired result. Moreover, when mapping, measurement areas are selected to a certain extent randomly. However, we have added a number of new points, so that we have something close to mapping the area.

Comments on the Quality of English Language

Not English, but the way authors describe results should be greatly improved. Their results are too poor quality anyway.

A: The authors have made significant changes to the manuscript and hope that the updated form will make it more suitable for publication. The text has been edited using professional proofreading (Cambridge Proofreading).

Reviewer 5 Report

Comments and Suggestions for Authors

This paper studies the three-layer structure of Ti-TiN-(Y,Ti,Al)N nanolayer coating.

Overall, it has certain scientific research value, but there are still some problems that need to be corrected:

1. The picture title should be centered. Obviously, most titles are not, and the image titles in the second half of the article are longer and should be streamlined appropriately.

2. The introduction does not need to be written separately, the content should be a whole.

3. The image text format should be unified into "Times New Roman".

4. "." should use the proportion of the marked element to represent the decimal point instead of ",".

5. The conclusion should be further condensed. The third conclusion is too broad and should include a quantitative description of the changes. The fourth conclusion is missing a period at the end.

Author Response

Reviewer 5 Comments and Suggestions for Authors

This paper studies the three-layer structure of Ti-TiN-(Y,Ti,Al)N nanolayer coating.

A: The authors are grateful to the Reviewer for his valuable assistance in improving the manuscript.

Overall, it has certain scientific research value, but there are still some problems that need to be corrected:

  1. The picture title should be centered. Obviously, most titles are not, and the image titles in the second half of the article are longer and should be streamlined appropriately.

A: Figure captions were centered on all Figures.

  1. The introduction does not need to be written separately, the content should be a whole.

A: The authors have made several changes to the Introduction to improve its quality and motivate research.

  1. The image text format should be unified into "Times New Roman".

A: Unfortunately, when obtaining a number of images, images automatically generated by programs were used, which do not have the ability to use Times New Roman.

  1. "." should use the proportion of the marked element to represent the decimal point instead of ",".

A: When obtaining the graphs, Excel was used; in the Cyrillic version, the use of “,” and not “.”was automatically set. Wherever possible, the authors used “.”

  1. The conclusion should be further condensed. The third conclusion is too broad and should include a quantitative description of the changes.

A: The third paragraph of the Conclusion has been supplemented in accordance with the Reviewer’s recommendations.

The fourth conclusion is missing a period at the end.

A: Typo corrected

Round 2

Reviewer 1 Report

Comments and Suggestions for Authors

Comparison with other tranditional coatings are suggested to be added at the end of the dicussion part to highlight the contribution of this work.

Author Response

Reviewer 1

A: The authors are grateful to the Reviewer for his invaluable assistance in improving the manuscript.

R: Comparison with other tranditional coatings are suggested to be added at the end of the dicussion part to highlight the contribution of this work.

A: Since in this work a comparison was made only with one of the widely used commercial coatings, it is noted in the Conclusion that

  1. When turning 1045 steel, a coating with 30 at. % yttrium showed better wear re-sistance compared to a commercial (Ti,Cr,Al) N coating. The coating with 63 at. % yttrium did not show an increase in wear resistance compared to the uncoated sample.

Comparison with other coatings was not carried out within the framework of this work, and comparison with the results presented in other works is not entirely correct, since test conditions may differ. However, the authors agree with the reviewer that a broader comparison is worth making, which will certainly be done in future papers.

Reviewer 3 Report

Comments and Suggestions for Authors

The manuscripte has undergone significant revisions. All the questions were answered very well.

Author Response

Reviewer 3

R: The manuscripte has undergone significant revisions. All the questions were answered very well.

A: The authors are grateful to the Reviewer for his invaluable assistance in improving the manuscript.

Reviewer 4 Report

Comments and Suggestions for Authors

Authors are still self-citating at a high volume of 20% or more, which is well above the journal's acceptable level. Unless authors reduce their self-citations (as well as group citations) to 5% or less, they should be automatically rejected.

I believe, [22], [23], [44]. [45], [46], [47], [49], [50], [51], [52], [53], [54], [56], [57], [58], [59], [60] are authors' group publications, as Grigoriev S, and Grigoriev S N are the same, as well as Volosova M and Volosova M.A. are the same, as they are from the same institution, Department of High-Efficiency Processing Technologies, Moscow State University of Technology.

Author Response

Reviewer 4

A: The authors are grateful to the Reviewer for his invaluable assistance in improving the manuscript.

R: Authors are still self-citating at a high volume of 20% or more, which is well above the journal's acceptable level. Unless authors reduce their self-citations (as well as group citations) to 5% or less, they should be automatically rejected.

I believe, [22], [23], [44]. [45], [46], [47], [49], [50], [51], [52], [53], [54], [56], [57], [58], [59], [60] are authors' group publications, as Grigoriev S, and Grigoriev S N are the same, as well as Volosova M and Volosova M.A. are the same, as they are from the same institution, Department of High-Efficiency Processing Technologies, Moscow State University of Technology.

A: Dear Mr. Reviewer,

Of course, the author agrees that excessive self-citation is unacceptable.

Since my scientific structure is quite large (more than 100 researchers, including more than 20 professors) and all the researchers work in the direction of modifying coatings, I think there is nothing strange in the fact that I refer to the works of Professor Grigoriev, who is a leading researcher in this field. The journal's rules do not limit references to the work of colleagues or former co-authors. References to such works are not self-citations. However, taking into account the Reviewer’s recommendations, I have significantly reduced the number of references to the works of Grigoriev and his colleagues.

I also reduced the number of self-citations and now there are 8 references out of 77, which is almost 10%.

Round 3

Reviewer 4 Report

Comments and Suggestions for Authors

Simply reduce the self-citations to less than 5%. I still find several numbers of self-citations could be reduced. 

Comments on the Quality of English Language

N/A

Author Response

Reviewer 4 Comments and Suggestions for Authors

R: Simply reduce the self-citations to less than 5%. I still find several numbers of self-citations could be reduced.

A: Self-citations are reduced to a minimum. Only 4 references left.